# Effect of Mild Alkali Treatment on the Structure and Physicochemical Properties of Normal and Waxy Rice Starches

**DOI:** 10.3390/foods13152449

**Published:** 2024-08-02

**Authors:** Zekun Xu, Xiaoning Liu, Chuangchuang Zhang, Mengting Ma, Zhongquan Sui, Harold Corke

**Affiliations:** 1Department of Food Science and Technology, School of Agriculture and Biology, Shanghai Jiao Tong University, Shanghai 200240, China; xuzekun@sjtu.edu.cn (Z.X.); lxn9520@sjtu.edu.cn (X.L.); cczhang@sjtu.edu.cn (C.Z.); 2Shanghai Jiao Tong University Sichuan Research Institute, Chengdu 610218, China; 3Department of Biotechnology and Food Engineering, Guangdong Technion—Israel Institute of Technology, Shantou 515063, China; harold.corke@gtiit.edu.cn; 4Faculty of Biotechnology and Food Engineering, Technion—Israel Institute of Technology, Haifa 3200003, Israel

**Keywords:** alkali treatment, hydrothermal treatment, pasting properties, rice starch

## Abstract

Mild alkali treatment can potentially be developed as a greener alternative to the traditional alkali treatment of starch, but the effect of mild alkali on starch is still understudied. Normal and waxy rice starches were subjected to mild alkali combined with hydrothermal treatment to investigate their changes in physicochemical properties. After mild alkali treatment, the protein content of normal and waxy rice starches decreased from 0.76% to 0.23% and from 0.89% to 0.23%, respectively. Mild alkali treatment decreased gelatinization temperature but increased the swelling power and solubility of both starches. Mild alkali treatment also increased the gelatinization enthalpy of waxy rice starch from 20.01 J/g to 25.04 J/g. Mild alkali treatment at room temperature increased the pasting viscosities of both normal and waxy rice starches, whereas at high temperature, it decreased pasting viscosities during hydrothermal treatment. Alkali treatment significantly changed the properties of normal and waxy rice starch by the ionization of hydroxyl groups and the removal of starch granule-associated proteins. Hydrothermal conditions promoted the effect of alkali. The combination of hydrothermal and alkali treatment led to greater changes in starch properties.

## 1. Introduction

Rice (*Oryza sativa* L.) is a staple food for half of the world population. Starch is the main component of rice grains, accounting for approximately 90% of the total constituents. Compared to other common starches, rice starch has some unique properties such as small size, non-allergenicity, and smooth and creamy characteristics. It is widely utilized in food and pharmaceutical application [1,2]. However, the uses of rice starch are limited by its instability under various temperature, pH, and shear condition [3]. To enhance the functional properties, rice starches are commonly modified to alter their properties, such as to decrease retrogradation tendency, increase freeze–thaw stability, and increase emulsifying ability [3,4].

Alkali is an effective method in changing the properties of starch and is widely used to enhance the desirable quality of some starch-based foods, e.g., maize tortilla and yellow alkaline noodle [5]. The effect of alkali on starch is varied and dependent on many factors, e.g., alkali reagent, concentration, time, and starch type [6,7,8]. Alkaline solution (3 M) is used to prepare cold water-soluble starch in the food industry [9]. Furthermore, alkaline soaking (>1% NaOH) alters the properties of starch by reducing the stability of starch granules [6,10,11]. However, high concentration of alkali is harmful to the environment. Recently, the demand for green processing is rising for a sustainable future. It is necessary to develop more mild (pH 8–12) alkali treatment at low concentrations (e.g., <0.1%) for starch processing in the industry.

Many research studies report that high pH (>13) can destroy the crystalline and amorphous region and even induce the gelatinization of starch at room temperature [6,10,12,13]. However, low-concentration alkali (0.1% NaOH solution, pH 12.4) has no significant effect on starch crystallinity, but significantly affects its pasting and thermal properties [11,14,15]. This indicates that mild alkali cannot destroy a crystal structure but may change the amorphous region to affect the properties of starch. Alkali can induce the ionization of hydroxyl groups in the amorphous region, which may promote the expansion of the starch granule and increase the flexibility of starch molecules by the repulsion of anions [16]. Therefore, the starch molecules may move and rearrange in the amorphous region, resulting in changes in granular structure and properties of starch.

Although the process of starch extraction removes a majority of proteins from starch, there are still small amounts of protein residing at the surface and/or interior the starch granule, called starch granule-associated proteins (SGAPs). SGAPs mainly comprise proteins related to starch synthesis, e.g., starch synthase (SS), granule-bound starch synthase (GBSS), 1,4-α-glucan-branching enzyme (GBE), and the residual seed storage proteins can be considered as SGAPs technologically. SGAPs can interact with starch molecules in various ways, including covalent bonding, hydrogen bonding, hydrophobic action, and van der Waal’s forces [17], affecting the properties of starch [18,19]. The presence of SGAPs can increase starch stability and inhibit starch swelling and gelatinization [17,20]. The removal of SGAPs weakens the granular structure of starch, resulting in the increase in swelling power and decrease in gelatinization temperature. Specially, GBSS, which is responsible for the synthesis of amylose, shows a greater effect on starch properties than other proteins, as the removal of GBSS specifically leads to an increase in the breakdown of starch. It is well known that proteins are sensitive to changes in pH. Alkali (>pH 12.5) can reduce SGAPs and change the distribution of SGAPs in starch granules [21]. Mild alkali may degrade SGAPs and loosen the starch granule to swell more easily, which reduces starch granule stability and changes the properties of the starch.

Mild alkali is also commonly used in starch chemical modification, sometimes combined with hydrothermal conditions (40–50 °C). It is well known that temperature is related to the molecular activity and reaction rate [22]. Thus, hydrothermal conditions can be used in mild alkali treatment, which can substantially increase molecular activity, promoting its effect. Furthermore, hydrothermal treatment at 40–50 °C with excess water is a physical method to alter the properties of starch (i.e., annealing) [23,24]. The combination of mild alkali and hydrothermal treatment might induce greater changes in starch properties.

Therefore, we hypothesized that mild alkali could influence starch properties by the ionization of hydroxy groups and the removal of SGAPs, while hydrothermal conditions might promote the effect of alkali. In this study, normal and waxy rice starches were treated under different mild alkaline conditions to investigate the effect on starch structure and properties. This study might be helpful in the development of greener alkali treatments for starch processing and provide new insights for understanding the mechanism of mild alkali in starch modification.

## 2. Materials and Methods

### 2.1. Materials

Normal rice and waxy rice were provided by Luzhou Rice Research Institute (Sichuan, China) with amylose contents of 11.6% and 0.82%, respectively. All chemicals and reagents used were of analytical grade.

### 2.2. Rice Starch Extraction

Starch was isolated following the method of Zhu et al. [25] with minor modifications. Rice flour (200 g) was steeped with 1 L 0.45% sodium metabisulfite solution at 4 °C for 24 h. After soaking, the rice flour was washed and milled with 1 L (0.3%, *w*/*w*) NaCl solution in a food blender and then passed through a 325-mesh sieve. After centrifugation at 3000× *g* for 10 min, the supernatant and the upper yellow protein layer were discarded. The white layer was resuspended with deionized water and centrifuged again. The washing step was repeated five times. The starch was filtered by a Büchner funnel and dried in an oven at 40 °C overnight.

### 2.3. Alkali Treatment

Alkali-treated starches were obtained by combinations of different pH (8.5, 9.9, and 11.3) and time–temperature conditions (at 25 °C for 1 h, named as room temperature condition, RTC, and at 50 °C for 18 h, named as high temperature condition, HTC). The starch suspension (35%, *w*/*v*) was magnetically stirred at 25 °C for 1 h (or at 50 °C for 18 h) and the pH was kept at 8.5 (or 9.9, 11.3) by an automatic titrator with 0.1 M NaOH. Then, starch suspension was adjusted to pH 6.5 and centrifuged at 3000× *g* for 15 min. Sediment was resuspended with distilled water and centrifuged again. This step was repeated three times to remove salt in the starch suspension. Starch was filtered through a Büchner funnel and dried at 40 °C for 24 h. Native starch and starch only treated at 50 °C for 18 h without alkali were set as control groups.

### 2.4. Starch Granule-Associated Protein Content

Starch granule-associated protein content was determined using the approved method 46-11A of AACC [26].

### 2.5. X-ray Diffraction

X-ray diffraction patterns of starches were collected using an X-ray diffractometer (Miniflex600, Rigaku Corporation, Tokyo, Japan) at 40 kV and 15 mA. The angular range of 2θ was 5–40° with 0.01° step interval at the rate of 2°/min. Crystalline peaks were analyzed and relative crystallinities (RCs) were calculated by MDI Jade 6.5 software (Materials Data, Livermore, CA, USA).

### 2.6. Swelling Power and Solubility

Swelling power (SP) and solubility (SOL) were determined following a previous method with minor modifications [27]. Starch (0.5 g, d.b.) and distilled water (40 mL) were mixed in a centrifuge tube by vortex and incubated at 85 °C for 30 min with a water bath. After cooling to room temperature, starch gel was centrifuged at 1800× *g* for 15 min. The supernatant was collected and dried at 150 °C to a constant weight. SP was represented as the ratio of the mass of wet sedimented gel to the dry mass of the gel. The SOL was represented as the percentage of dried supernatant solid mass based on the dry mass of starch sample.

### 2.7. Thermal Properties

Thermal properties were determined with a differential scanning calorimeter (DSC 2500, TA Instruments, New Castle, DE, USA). Starch (2 mg, d.b.) was mixed with distilled water (6 mL) in an aluminum DSC pan. The pan was sealed and equilibrated at room temperature for 24 h. The sample was scanned from 30 °C to 120 °C at a heating rate of 10 °C/min. Gelatinization enthalpy (Δ*H*), onset (T_o_), peak (T_p_), conclusion temperature (T_c_), and gelatinization temperature range (T_c_–T_o_) were calculated using TRIOS 4.2 software (TA Instruments, New Castle, DE, USA).

### 2.8. Pasting Properties

The pasting parameters were determined using a RVA4500 Rapid Visco Analyzer (Perten Instruments, Stockholm, Sweden) according to a previous method [20]. Starch slurry (28 g, 7%) was added into the RVA container. The sample was (1) held at 50 °C for 60 s, (2) heated to 95 °C in 222 s, (3) held at 95 °C for 150 s, (4) cooled to 50 °C in 222 s, (5) held at 50 °C for 120 s. Peak viscosity (PV), trough viscosity (TV), final viscosity (FV), breakdown (BD), and setback (SB) were obtained.

### 2.9. Statistical Analysis

Analysis of variance (ANOVA) was performed to compare treatment means. The results were evaluated by Duncan’s test using the SPSS software (SPSS 24.0, IBM, Armonk, NY, USA) and *p* < 0.05 was considered as significant difference.

## 3. Results and Discussion

### 3.1. SGAP Content

The SGAP contents of both normal and waxy rice starch significantly decreased as a function of pH (Table 1). This was consistent with a previous report stating that alkali decreased the protein contents of rice starch [28]. The irreversible effect of alkali can partly remove or degrade SGAPs, resulting in the reduction in protein content [28]. The reduction in protein contents for normal and waxy rice starches in RTCs was found to be from 0.73% to 0.66% and from 0.96% to 0.85%, respectively. The insolubility of rice proteins is mainly due to the presence of sulfhydryl groups from cysteine residues and amide bonds from glutamine and asparagine residues, which could form disulfide bonds and hydrogen bonds to maintain the protein structure [29]. The cleavage of disulfide bond and deamination of amide bonds could occur in an alkaline condition, increasing the solubility of SGAPs [8,30]. Furthermore, the starch granules swell in alkali; thus, SGAPs could leach out more easily [28]. The reduction in proteins (around 10% for normal rice starch and 12% for waxy rice starch) was much lower than the previously reported protein reduction (up to 50%) for wheat starch under alkali treatment (0.025 M NaOH for 3 weeks) [14]. It indicated that the disruption of SGAPs during alkali treatment at pH < 11.3 was weak, and the very low concentration of alkali could only remove little SGAPs from starch granules.

There was no significant difference in the SGAP content between native starch and their counterpart in HTCs, indicating hydrothermal treatment did not affect the protein content. Normal and waxy rice starches in HTCs showed greater reduction in protein content than their counterparts in RTCs during alkali treatment, suggesting that hydrothermal effect could promote the destructive effect of alkali for SGAPs. In particular, the protein contents of normal and waxy rice starches in HTC-11.3 both decreased to 0.23%. The higher temperature could increase the reactivity of alkali molecules [22], resulting in a significant reduction in protein content. It was reported that an increase in temperature from 25 °C to 55 °C during maize starch extraction by 0.1% NaOH reduced the protein content of starch from 0.75% to 0.19% [31], similar to the result of this study. It might be because high temperature could promote the destruction of disulfide bonds and the deamination reaction in alkaline conditions, thus alkali treatment could efficiently destroy the sulfhydryl groups and induce the deamination of glutamine and asparagine residues [30]. Therefore, the structure of SGAPs was destroyed and the solubility of SGAPs increased, resulting in the great reduction in protein content in HTC-11.3.

### 3.2. Crystalline Structure

Both normal and waxy rice starches showed diffraction peaks at 15°, 17°, 18° and 23° (Figure 1), which was typical A-type starch. After alkali and hydrothermal treatment, the positions of peaks remained constant, indicating that these treatments did not induce a significant change in the crystal type of rice starch [32]. There was no significant difference in relative crystallinity (RC) between native starch and their counterpart in HTCs. The conditions for hydrothermal treatment in this study were similar to those for annealing [24]. In this study, the hydrothermal treatment had no effect on the RCs of starches, consistent with the result of previous research in wheat starch [32].

The alkali treatment had no effect on the RCs of starch, indicating that alkali treatment did not destroy the crystal structure, as the low concentration of alkali was insufficient to disturb the closely packed double helix. It was reported that 0.1 M NaOH solution (pH 13, 25 °C) induced the expansion of amorphous regions but had no significant effect on crystal regions [33]. And it was believed that alkali environment in chemical modifications (pH 8–12, 25 °C–50 °C) could induce the ionization of starch hydroxyl group by disrupting hydrogen bond of chains, causing reversible swelling of starch granule by the repulsion of anions [16,33]. Starch hydrogen groups were reinforced by the hydrogen bond of double-helical structure in crystalline regions during alkali treatment. Therefore, no changes in starch RCs occurred with alkali treatment, similar to the result of alkali-treated sago starch (pH 12.4, 25 °C) [11].

### 3.3. Swelling Power and Solubility

In RTC, the SP of normal and waxy rice starches increased by 14.3% and 14.0%, respectively, as a function of pH (Table 1). The ionization of hydroxy groups by alkali accounted for the increased SP in RTC. The repulsion of anions could promote the movement of starch molecules and increase intermolecular distance. Thus, water could enter more easily [16], resulting in the increase in SP. Furthermore, it is believed that SGAPs can maintain starch granular structure and inhibit granule swelling [20]. The partial removal of SGAPs during alkali treatment increased the SP of starch to some extent.

There was no difference in SP between the native normal rice starch and its counterpart in HTCs, while the native waxy rice starch with HTC showed a higher SP than the native waxy starch. The SP of normal and waxy rice starches in HTCs increased by 13.0% and 9.4%, respectively. The increased degree of SP by alkali treatment for both normal and waxy rice starches in HTCs was slightly lower than the increase for the starch in RTCs, indicating the hydrothermal treatment did not promote the effect of alkali on SP, this was similar to a previous study [32]. In particular, the SP of normal rice starch with HTC-11.3 increased from 11.4 g/g to 22.9 g/g, whereas the SP of waxy rice starch with HTC-11.3 only slightly increased from 27.7 g/g to 33.3 g/g. The difference of SP between normal and waxy rice starches with HTC-11.3 might be related to a special type of SGAPs in normal rice starch, granule-bound starch synthase (GBSS). GBSS is a protein responsible for the synthesis of amylose [34]. GBSS is tightly bound to starch compared to other SGAPs. It was reported that the removal of GBSS led to a greater reduction in starch stability [35,36]. Therefore, when the reduction ratio of SGAPs was similar, the increase in SP of normal rice starch was greater than that of waxy rice starch.

In RTC, the alkali treatment slightly increased the SOL of starches. It could be attributed to the ionization of hydroxy groups by alkali, which increased the mobility of starch molecules, and thus making them more easily leach out [14]. Hydrothermal treatment in HTC decreased SOL values of native normal and waxy rice starches from 4.1% to 3.6% and from 12.5% to 10.5%, respectively. The hydrothermal treatment in HTC was similar to annealing, which was believed to enhance the stability of starch granule [23]. However, the SOL of starches in HTC increased as a function of pH, which indicated that hydrothermal condition promoted the destructive influence of alkali treatment on the starch granule. Additionally, the SOL of normal starch in HTC-11.3 significantly increased from 3.6% to 10.8%, similar to the change in SP of normal rice starch in HTC-11.3.

### 3.4. Thermal Properties

Compared to native starch in RTC, native starch in HTC showed higher gelatinization temperature (T_o_, T_p_) and narrower temperature range (Table 2). It could be attributed to the annealing effect, which could reorient and rearrange the starch molecules, optimizing the crystal structure of starch [24,37]. As the result, the stability of starch would increase, thus increasing its gelatinization temperature [24]. The native waxy rice starch in HTC also showed higher Δ*H*, indicating that the rearrangement of starch molecules could induce the formation of hydrogen bonds and double-helical structures [24]. However, no similar phenomenon was observed in native normal rice starches in HTC. This indicated that the presence of amylose might inhibit the rearrangement of amylopectin chains and prevent the formation of new hydrogen bonds among amylopectin chains. Amylose is believed to mainly exist in amorphous regions, and can intersperse between amylopectin chains in crystal regions, and even cross the crystal regions [38]. Due to the ultra-long chains of amylose and the entanglement with amylopectin in crystal regions, the mobility of amylose might be restricted in the limited space. The presence of amylose might entangle with amylopectin chains and prevent amylopectin from arrangement.

The T_o_ and T_p_ of both normal and waxy rice starches were slightly reduced after alkali treatment, indicating that the stability of the crystalline region was reduced, which might be related to the decrease in SGAPs. The presence of SGAPs can maintain the structure of starch granule and enhance the stability of starch granule [39]. It was reported that the removal of SGAPs can significantly decrease the gelatinization temperature of starches [40]; this might be because SGAPs act as a barrier to hinder water from entering the starch granule, thus enhancing the thermal resistance of starch and further delaying the gelatinization [40]. It could be observed that different pH of the alkali treatment showed little effect on the T_o_ and T_p_ of starches in RTCs but had a significant effect on those in HTCs, i.e., the higher pH led to greater reduction in gelatinization temperature. This result indicated that hydrothermal treatment might promote the effect of alkali, weakening the stability of starches.

As shown in Table 2, it was worth noting that T_c_ and Δ*H* of waxy rice starches in HTC significantly increased as a function of pH, which might be related to the ionization of hydroxyl groups. Despite the destruction of amorphous regions, the ionization of hydroxy groups induced the free hydroxyl group of the amylopectin side chain in the amorphous region of starch to dissociate into anions [12]. The intermolecular repulsive force between anions promotes the flexibility of side chains of amylopectin and the flexible chains can move and reorientate [16]. When the starch slurries were neutralized after alkali treatment, hydroxyl groups were reformed by anions and the free hydroxyl group of the arranged side chains might form new hydrogen bonds, resulting in the increase in T_c_ and Δ*H*. But alkali treatment had no significant effect on the T_c_ and Δ*H* of normal rice starch, which was similar to the effect of annealing on enthalpy [23]. The presence of amylose might inhibit the flexibility of amylopectin side chains and prevent the chain rearrangement in hydrothermal and alkaline treatment. Therefore, the enthalpy of normal rice starch changed slightly. These results indicated that the presence of amylose in starch played an important role in alkali and hydrothermal treatment.

### 3.5. Pasting Properties

The alkali treatment in RTCs and HTCs showed different effects on the pasting properties of starches (Table 3). Normal rice starch showed higher PV, TV, and FV, compared to waxy rice starch. In RTCs, PV, TV, and FV of waxy rice starches increased first and then decreased as a function of pH. Meanwhile, the alkali treatment in RTC increased the TV and FV of normal rice starches, but decreased their BD. The alkali treatment could induce the formation of hydroxy anions. The repulsion of anions could enlarge the interspace to let water molecules enter more easily, thus increasing the viscosity [5].

However, in HTC groups, alkali treatment significantly decreased the pasting viscosities of both normal and waxy rice starches, which was similar to the result of long-term alkali treatment [5,10]. The stronger and long-term alkali treatment enhanced by hydrothermal effect in HTCs would cause greater destruction at the amorphous regions, weakening the granular structure [10]. The reduction in SGAP content might be another reason for the decreased viscosity of normal and waxy rice starches in HTC alkali treatment. PV, BD, and FV of small-granule starches decreased after the removal of SGAPs [41], indicating a weaker granular structure and a greater tendency to lose viscosity under shear. During pasting, the weak structure of the starch granule in HTC would be disrupted before it reached its maximum swelling capacity, leading to a decrease in viscosity. The normal rice starch with HTC-11.3 showed greater decrease in TV and FV, mainly due to the reduction in SGAPs (particular GBSS). The reduction in SGAPs significantly reduced the granular stability. Starch granules could swell and rupture more easily, resulting in the decrease in trough and final viscosity [36].

## 4. Conclusions

Alkali treatment significantly decreased the SGAP content and gelatinization temperature, but increased the swelling power and solubility of both normal and waxy rice starches. Alkali treatment also increased the gelatinization enthalpy of waxy rice starches but showed little effect on gelatinization enthalpy of normal rice starch. Alkali treatment in RTC increased the pasting viscosities of both normal and waxy rice starches, whereas alkali treatment in HTC decreased pasting viscosities of starches. The results showed that alkali treatment had little effect on the crystalline structure of starch, but significantly changed the physicochemical properties of normal and waxy rice starch mainly by the removal of SGAPs and the ionization of hydroxyl groups. The rearrangement of starch molecules might occur during alkali treatment by ionization of hydroxy groups. The hydrothermal treatment would enhance the effect of alkali treatment, leading to greater changes in starch properties. The reduction in SGAPs by alkali reduced the granular stability, leading to the starch rupture more easily. It is undesirable for some food applications as a bulking agent, sorbent, etc. The different changes comparing normal and waxy rice starches also indicated that the amylose content significantly impacted the effect of alkali treatment.

## Figures and Tables

**Figure 1 foods-13-02449-f001:**
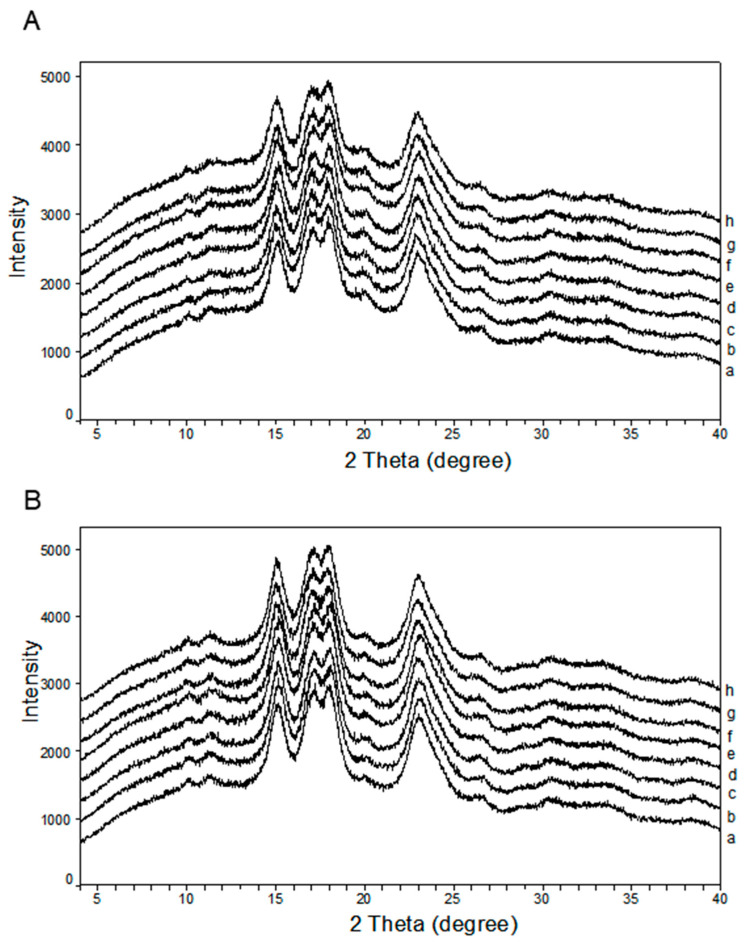
X-ray diffraction patterns of (**A**) normal rice starch and (**B**) waxy rice starch: (a) native, (b) RTC-8.5, (c) RTC-9.9, (d) RTC-11.3, (e) HTC-native, (f) HTC-8.5, (g) HTC-9.9, and (h) HTC-11.3.

**Table 1 foods-13-02449-t001:** Protein content, relative crystallinity, swelling power, and solubility of starches with different treatments.

Sample		Alakli Treatment	Protein Content %	RC %	SP (g/g)	SOL %
Normal rice	native	-	0.76 ± 0.02 ab	22.0 ± 0.5 a	11.4 ± 0.1 e	4.1 ± 0.1 de
	RTC-8.5	25 °C 1 h pH 8.5	0.73 ± 0.01 bc	22.0 ± 0.4 a	13.1 ± 0.4 b	6.7 ± 0.9 b
	RTC-9.9	25 °C 1 h pH 9.9	0.75 ± 0.01 abc	22.1 ± 0.4 a	12.0 ± 0.1 de	4.2 ± 0.0 de
	RTC-11.3	25 °C 1 h pH 11.3	0.66 ± 0.04 d	22.0 ± 0.6 a	12.4 ± 0.4 cd	4.7 ± 0.7 cde
	HTC-native	50 °C 18 h	0.79 ± 0.01 a	22.2 ± 0.3 a	11.4 ± 0.2 e	3.6 ± 0.2 e
	HTC-8.5	50 °C 18 h pH 8.5	0.74 ± 0.04 abc	21.9 ± 0.9 a	12.7 ± 0.0 bc	5.3 ± 0.2 c
	HTC-9.9	50 °C 18 h pH 9.9	0.70 ± 0.04 cd	22.1 ± 0.3 a	12.9 ± 0.1 bc	4.9 ± 0.0 cd
	HTC-11.3	50 °C 18 h pH 11.3	0.23 ± 0.01 e	21.8 ± 0.6 a	22.9 ± 3.4 a	10.8 ± 4.0 a
Waxy rice	native	-	0.96 ± 0.04 a	26.5 ± 0.2 ab	27.7 ± 1.2 ef	12.5 ± 1.3 b
	RTC-8.5	25 °C 1 h pH 8.5	0.89 ± 0.01 abc	26.6 ± 0.2 ab	31.6 ± 0.1 abc	13.5 ± 0.0 b
	RTC-9.9	25 °C 1 h pH 9.9	0.88 ± 0.01 abc	27.7 ± 0.1 a	27.6 ± 0.4 f	12.1 ± 0.2 bc
	RTC-11.3	25 °C 1 h pH 11.3	0.85 ± 0.01 bc	26.5 ± 0.1 ab	29.5 ± 0.8 de	13.4 ± 0.0 b
	HTC-native	50 °C 18 h	0.94 ± 0.07 ab	26.6 ± 0.4 ab	30.5 ± 0.6 bcd	10.5 ± 0.5 c
	HTC-8.5	50 °C 18 h pH 8.5	0.81 ± 0.01 c	25.7 ± 0.0 bc	29.8 ± 1.5 cd	15.6 ± 0.3 a
	HTC-9.9	50 °C 18 h pH 9.9	0.91 ± 0.01 ab	24.5 ± 1.5 c	31.8 ± 0.3 ab	16.2 ± 0.5 a
	HTC-11.3	50 °C 18 h pH 11.3	0.23 ± 0.05 d	26.4 ± 1.0 ab	33.3 ± 0.1 a	15.9 ± 1.4 a

Data from three technical replications were presented as mean ± standard deviation. Values in each column (within the same cultivar) with different letters are significantly different (*p* < 0.05) according to Duncan’s test.

**Table 2 foods-13-02449-t002:** Thermal properties of starches with different treatments.

Sample		Δ*H* (J/g)	T_o_ (°C)	T_p_ (°C)	T_c_ (°C)	T_c_–T_p_ (°C)
Normal rice	native	11.25 ± 0.39 ab	63.89 ± 0.01 e	69.95 ± 0.04 c	75.77 ± 0.15 ab	11.88 ± 0.14 a
	RTC-8.5	12.81 ± 0.98 a	63.40 ± 0.06 f	69.51 ± 0.11 d	75.55 ± 0.05 bc	12.15 ± 0.06 a
	RTC-9.9	11.43 ± 0.52 ab	63.43 ± 0.02 f	69.69 ± 0.06 cd	75.37 ± 0.12 c	11.94 ± 0.14 a
	RTC-11.3	10.86 ± 0.6 ab	63.47 ± 0.08 f	69.56 ± 0.10 d	75.45 ± 0.13 bc	11.97 ± 0.09 a
	HTC-native	10.72 ± 0.54 b	66.78 ± 0.04 a	70.98 ± 0.13 a	75.94 ± 0.16 a	9.16 ± 0.18 e
	HTC-8.5	11.02 ± 0.48 ab	65.97 ± 0.07 b	70.71 ± 0.11 b	75.69 ± 0.12 abc	9.72 ± 0.04 d
	HTC-9.9	11.18 ± 1.60 ab	65.63 ± 0.15 c	70.60 ± 0.23 b	75.73 ± 0.25 abc	10.10 ± 0.18 c
	HTC-11.3	12.47 ± 1.25 ab	64.65 ± 0.10 d	69.93 ± 0.20 c	75.57 ± 0.31 bc	10.91 ± 0.38 b
Waxy rice	native	20.01 ± 1.68 c	74.28 ± 0.20 b	79.88 ± 0.08 bc	87.13 ± 0.54 d	12.85 ± 0.68 d
	RTC-8.5	21.43 ± 2.15 bc	73.75 ± 0.35 c	79.65 ± 0.32 cd	88.71 ± 1.21 c	14.96 ± 1.49 bc
	RTC-9.9	21.18 ± 2.9 bc	73.64 ± 0.17 c	79.72 ± 0.08 bcd	87.38 ± 1.10 d	13.74 ± 1.21 cd
	RTC-11.3	21.86 ± 2.38 bc	73.75 ± 0.12 c	79.51 ± 0.08 de	87.49 ± 0.60 d	13.75 ± 0.63 cd
	HTC-native	23.12 ± 0.47 abc	74.90 ± 0.11 a	80.15 ± 0.12 a	85.55 ± 0.43 e	10.65 ± 0.43 e
	HTC-8.5	22.84 ± 2.70 bc	74.16 ± 0.26 b	82.55 ± 5.44 ab	87.23 ± 4.42 bc	13.06 ± 4.30 b
	HTC-9.9	23.76 ± 1.69 ab	73.30 ± 0.23 d	79.94 ± 0.17 ab	90.05 ± 0.94 ab	16.75 ± 1.15 a
	HTC-11.3	25.04 ± 0.70 a	73.33 ± 0.14 d	79.34 ± 0.15 e	91.03 ± 0.87 a	17.70 ± 0.84 a

Data from three technical replications were presented as mean ± standard deviation. Values in each column (within the same cultivar) with different letters are significantly different (*p* < 0.05) according to Duncan’s test.

**Table 3 foods-13-02449-t003:** Pasting properties of starches with different treatments.

Sample		PV (mPa·s)	TV (mPa·s)	BD (mPa·s)	FV (mPa·s)	SB (mPa·s)
Normal rice	native	1639 ± 15 ab	1223 ± 11 b	416 ± 3 b	1802 ± 6 c	579 ± 5 bc
	RTC-8.5	1658 ± 5 a	1412 ± 11 a	246 ± 6 d	1949 ± 21 b	537 ± 10 d
	RTC-9.9	1626 ± 5 ab	1414 ± 1 a	212 ± 4 d	2010 ± 7 ab	595 ± 8 abc
	RTC-11.3	1616 ± 4 ab	1406 ± 5 a	209 ± 1 d	2026 ± 4 a	619 ± 1 ab
	HTC-native	1644 ± 31 b	1237 ± 25 b	406 ± 56 b	1800 ± 28 c	562 ± 3 cd
	HTC-8.5	1527 ± 62 ab	1208 ± 19 b	318 ± 43 c	1809 ± 27 c	600 ± 8 abc
	HTC-9.9	1490 ± 8 c	1121 ± 28 c	369 ± 20 bc	1747 ± 29 c	626 ± 1 a
	HTC-11.3	1590 ± 13 c	829 ± 8 d	760 ± 5 a	1250 ± 54 d	421 ± 47 e
Waxy rice	native	1171 ± 16 d	663 ± 3 e	508 ± 13 b	810 ± 4 d	146 ± 1 a
	RTC-8.5	1211 ± 8 c	720 ± 4 c	491 ± 4 b	844 ± 6 c	124 ± 1 d
	RTC-9.9	1251 ± 1 b	780 ± 0 b	471 ± 1 c	911 ± 2 b	131.5 ± 2 c
	RTC-11.3	1111 ± 11 e	673 ± 6 d	437 ± 5 de	813 ± 4 d	139 ± 2 b
	HTC-native	1539 ± 5 a	887 ± 3 a	652 ± 8 a	984 ± 3 a	97 ± 0 e
	HTC-8.5	1070 ± 10 f	623 ± 3 f	446 ± 6 d	746 ± 2 e	123 ± 1 d
	HTC-9.9	993 ± 11 g	588 ± 3 g	405 ± 7 f	723 ± 7 f	134 ± 3 c
	HTC-11.3	970 ± 10 h	543 ± 1 h	426 ± 9 e	667 ± 1 g	124 ± 0 d

Data from two technical replications were presented as mean ± standard deviation. Values in each column (within the same cultivar) with different letters are significantly different (*p* < 0.05) according to Duncan’s test.

## Data Availability

The original contributions presented in the study are included in the article, further inquiries can be directed to the corresponding authors.

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
