# Peer review of "Effect of Mild Alkali Treatment on the Structure and Physicochemical Properties of Normal and Waxy Rice Starches"

_foods, 2024, doi:10.3390/foods13152449_

Round 1
Reviewer 1 Report
Comments and Suggestions for Authors
Introduction:
Line 42: Some word is missing here
Line 69: The authors present the alkali effects on starch and proteins but do not mention fats. Typically, fats are oxidized by the use of alkali yielding a poor flavor to the rice starch used on extrusion products. Any comments about this?
Materials and Methods:
Line 98: any rationale for these conditions? Like previous works or experimental tests?
Line 98: I think it would be beneficial to make a table with the conditions. For example table 1 could have the information. Im sending my proposal
|
Sample |
Rice Starch |
Temperature |
pH |
Protein Content (%) |
|
Native |
Normal |
- |
- |
|
|
NRTC 8.5 |
Normal |
25 |
8.5 |
... |
|
NRTC 9.9 |
Normal |
25 |
9.9 |
|
|
NRTC 11.3 |
Normal |
25 |
11.3 |
|
|
NHTC - native |
Normal |
50 |
- |
|
|
NHTC 8.5 |
Normal |
50 |
8.5 |
|
|
NHTC 9.9 |
Normal |
50 |
9.9 |
|
|
NHTC 11.3 |
Normal |
50 |
11.3 |
|
|
Native |
waxy |
- |
- |
|
|
WRTC 8.5 |
waxy |
50 |
8.5 |
|
|
WRTC 9.9 |
waxy |
50 |
9.9 |
|
|
WRTC 11.3 |
waxy |
50 |
11.3 |
|
|
WHTC- native |
waxy |
50 |
- |
|
|
WHTC 8.5 |
waxy |
50 |
8.5 |
|
|
WHTC 9.9 |
waxy |
50 |
9.9 |
|
|
WHTC 11.3 |
waxy |
50 |
11.3 |
|
Results and discussion
Line 158: Please provide more detail on the role of SGAPs in starch functionality and how their reduction could impacts the overall properties of the starch.
Line 168: Please comment about potential denaturation of proteins and consequent increase in solubility
Line 174: Please provide figure 1
Line 184: please provide more detailed explanations on the molecular interactions within the crystalline regions that prevent significant of crystallinity, for instance, the treatment did not cause starch granules disruptions.
Line 228: the author mentions that the presence of amylose inhibits the rearrangement of amylopectin chains, preventing the formation of new hydrogen bonds. This is a critical point, so please discuss this in more depth. How does amylose interfere with these processes at the molecular level?
Line 233: Please comment on how does the reduction in SGAPs specifically affect the crystalline structure and stability of starch granules?
Line 240:
The increase in gelatinization enthalpy (ΔH) for waxy rice starch with alkali treatment is attributed to the rearrangement of starch molecules. Please expanded this explanation to describe the formation of new hydrogen bonds and double helix structures in more detail.
Line 261:
The increase in pasting viscosities (PV, TV, and FV) at room temperature conditions (RTC) is discussed, but the molecular reasons behind this increase need further elaboration. How do the ionization of hydroxyl groups and removal of SGAPs specifically affect the granular structure and water absorption capacity?
Line 280: please provide a clear comparison on how the presence of amylose influence the pasting properties compared to waxy starches under similar treatment conditions?
Table 1 and Table 2 – please correct the significant figures
Author Response
Comment 1: Introduction:
Line 42: Some word is missing here
Response 1: Thanks for your significant suggestions. We revised the manuscript according to the reviewer’s suggestion (Line 45).
Comment 2: Line 69: The authors present the alkali effects on starch and proteins but do not mention fats. Typically, fats are oxidized by the use of alkali yielding a poor flavor to the rice starch used on extrusion products. Any comments about this?
Response 2: Thanks for your significant suggestions. We carried out a preliminary experiment to determine the lipid content during alkali treatment. The results showed that the lipid contents in rice starch was significantly less than protein contents, indicating that there was no significant change during alkali treatment. For example, the lipid content of native rice starch and HTC-11.3 were 0.16±0.02% and 0.17±0.01%, respectively. In addition, the statistical analysis showed that there was no significant difference. Therefore, in this study we focused on the alkali effects on starch proteins but not lipids.
Comment 3: Materials and Methods:
Line 98: any rationale for these conditions? Like previous works or experimental tests?
Response 3: Thanks for your significant suggestions. The conditions in this study were referred to the alkali condition used in starch modification (25 – 50°C, 1 – 18 h). For example, the starch with acetic anhydride was with pH 8.5 at room temperature for 1 h, and the hydroxypropylation of starch was with pH 11.5 at 45°C for 18 h. We hope the study could be helpful for the understanding the effect of alkali treatment alone and/or in combination with other modifications.
Comment 4: Line 98: I think it would be beneficial to make a table with the conditions. For example table 1 could have the information. Im sending my proposal
Response 4: Thanks for your significant suggestions. We added the Table 1 following the reviewer’s suggestion (Line 177).
Comment 5: Results and discussion
Line 158: Please provide more detail on the role of SGAPs in starch functionality and how their reduction could impacts the overall properties of the starch.
Response 5: Thanks for your significant suggestions. We revised the manuscript according to the reviewer’s suggestion. The presence of SGAPs generally enhanced the integrity and rigidity of starch granules, and inhibited starch swelling and gelatinization [29,30], as it might warp with starch granule, and act as a barrier to starch gelatinization [30-32]. The reduction of SGAPs induced starch to swell and gelatinize more easily, resulting in the changes of swelling, pasting, and thermal properties (Line 162-166).
Comment 6: Line 168: Please comment about potential denaturation of proteins and consequent increase in solubility
Response 6: Thanks for your significant suggestions. We revised the manuscript according to the reviewer’s suggestion. higher temperature might induce the denaturation of partial thermal-sensitive proteins, increasing the solubility of SGAPs (Line 173-175)
Comment 7: Line 174: Please provide figure 1
Response 7: Thanks for your significant suggestions. We corrected this mistake and added the X-ray diffraction patterns as Figure.1 (Line 200)
Comment 8: Line 184: please provide more detailed explanations on the molecular interactions within the crystalline regions that prevent significant of crystallinity, for instance, the treatment did not cause starch granules disruptions.
Response 8: Thanks for your significant suggestions. We revised the manuscript according to the reviewer’s suggestion. The alkali treatment had no effect on the RC of starch, indicating that alkali treatment did not destroy the crystal structure, as the low concentration of alkali was insufficient to disturb the closely packed double helix. Alkali could induce ionization of starch hydroxyl group by disrupting hydrogen bond of chains, causing reversible swelling of starch granule by repulsion of anions [16,36]. Starch hydrogen groups were reinforced by the hydrogen bond of double helical structure in crystalline regions during alkali treatment. Therefore, no changes of starch RC occurred with alkali treatment (Line 192-198).
Comment 9: Line 228: the author mentions that the presence of amylose inhibits the rearrangement of amylopectin chains, preventing the formation of new hydrogen bonds. This is a critical point, so please discuss this in more depth. How does amylose interfere with these processes at the molecular level?
Response 9: Thanks for your significant suggestions. We revised the manuscript according to the reviewer’s suggestion. Amylose is believed to mainly exist in amorphous regions, and can interspersed be-tween amylopectin chains in crystal regions, and even cross the crystal regions [41]. Due to the ultra-long chains of amylose and the entanglement with amylopectin in crystal regions, the mobility of amylose might be restricted in the limited space. The presence of amylose might entangle with amylopectin chains and prevent amylopectin from arrangement (Line 245-250).
Comment 10: Line 233: Please comment on how does the reduction in SGAPs specifically affect the crystalline structure and stability of starch granules?
Response 10: Thanks for your significant suggestions. We revised the manuscript according to the reviewer’s suggestion. The presence of SGAPs generally enhanced the integrity and rigidity of starch granules, and inhibited starch swelling and gelatinization [29,30], as it might warp with starch granule, and act as a barrier to starch gelatinization [30-32] (Line 162-165).
Comment 11: Line 240:
The increase in gelatinization enthalpy (ΔH) for waxy rice starch with alkali treatment is attributed to the rearrangement of starch molecules. Please expanded this explanation to describe the formation of new hydrogen bonds and double helix structures in more detail.
Response 11: Thanks for your significant suggestions. We revised the manuscript according to the reviewer’s suggestions. The intermolecular repulsive force between anions promotes the flexibility of side chains of amylopectin and the flexible chains can move and reorientate [16]. When the starch slurries were neutralized after alkali treatment, hydroxyl groups were reformed by anions and the free hydroxyl group of the arranged side chains might form new hydrogen bonds (Line 264-268).
Comment 12: Line 261:
The increase in pasting viscosities (PV, TV, and FV) at room temperature conditions (RTC) is discussed, but the molecular reasons behind this increase need further elaboration. How do the ionization of hydroxyl groups and removal of SGAPs specifically affect the granular structure and water absorption capacity?
Response 12: Thanks for your significant suggestions. We revised the manuscript according to the reviewer’s suggestions. The alkali treatment could induce the formation of hydroxy anions. The repulsion of anions could enlarge the interspace to let water molecules enter more easily, leading increasing the viscosity [5] (Line 283-285).

Reviewer 2 Report
Comments and Suggestions for Authors
Modification of physicochemical properties of starches are attractive for food industry. Authors investigated mild alkali treatment on the structure and physicochemical properties of normal amylose and waxy rice starches. After mild alkali treatment, the protein contents (PGAPs) were drastically decreased in normal amylose and waxy rice starches. This treatment also resulted in gelatinization temperature and pasting properties, due to the removal of proteins ionization of hydroxyl groupes. Obtained results would help readers to improve the physicochemical properties of rice starches. However, reviewer raises some questions and comments.
Before reviewing, Figure 1 about x-ray diffraxtion patterns was not shown in this present manuscript. Therefore, the diffraction peak patterns (angle vs relative intensity) of samples couldn't be correctly evaluated (results and discussion 3.2 crystalline structure). No figure, No discussion. If some results have the specific peak or changes in diffraction pattern, authors must explain in detail.
Reduction of SGAPs by mild alkali treatment is benefical for normal amylose and waxy rice starches. What is the demerit of mild alkali modification? Although reviewer could understand the merit of this treatment, there were no explanation about demerit. It's better to add the texts.
3.3. Swelling power and solubility. Authors explained that the SP of normal amylose starch with HTC-11.3 increased (11.4 to 22.9), due to the great reduction of SGAPs content (0.76 to 0.23) in lines 206-208. In Table1, results of waxy rice showed much more reduction of SGAPs (0.96 to 0.23), however, the SP was slightly increased (27.7 to 33.3). What were the main reasons for these differences (relationships between SGAPs and SP) ? It is essential to add discussions with the references in 208.
Comments on the Quality of English Language
Modification of physicochemical properties of starches are attractive for food industry. Authors investigated mild alkali treatment on the structure and physicochemical properties of normal amylose and waxy rice starches. After mild alkali treatment, the protein contents were drastically decreased in normal amylose and waxy rice starches. This treatment also resulted in gelatinization temperature and pasting properties, due to the removal of proteins ionization of hydroxyl groupes. Obtained results would help readers to improve the physicochemical properties of rice starches. However, reviewer raises some questions and comments.
Before reviewing, the figure 1 about x-ray diffraxtion patterns were not shown in this present manuscript. Therefore, the diffraction peak patterns (angle vs relative intensity) of samples couldn't be correctly evaluated (results and discussion 3.2 crystalline structure). No figure, No discussion. If some results have the specific peak or changes in diffraction pattern, authors must explain in detail.
Reduction of SGAPs by mild alkali treatment is benefical for normal amylose and waxy rice starches. What is the demerit of mild alkali modification? Although reviewer could understand the merit of this treatment, there were no explanation of demerit. It's better to add the texts.
3.3. Swelling power and solubility Authors explained that the SP of normal amylose starch with HTC-11.3 increased (11.4 to 22.9), due to the great reduction of SGAPs content (0.76 to 0.23) in lines 206-208. In Table1, results of waxy rice showed much more reduction of SGAPs (0.96 to 0.23), however, the SP was slightly increased (27.7 to 33.3). What were the main reasons for these differences (relationships between SGAPs and SP) ? It is essential to add discussions with the references in 208.
Author Response
Comment 1: Modification of physicochemical properties of starches are attractive for food industry. Authors investigated mild alkali treatment on the structure and physicochemical properties of normal amylose and waxy rice starches. After mild alkali treatment, the protein contents (PGAPs) were drastically decreased in normal amylose and waxy rice starches. This treatment also resulted in gelatinization temperature and pasting properties, due to the removal of proteins ionization of hydroxyl groupes. Obtained results would help readers to improve the physicochemical properties of rice starches. However, reviewer raises some questions and comments.
Before reviewing, Figure 1 about x-ray diffraxtion patterns was not shown in this present manuscript. Therefore, the diffraction peak patterns (angle vs relative intensity) of samples couldn't be correctly evaluated (results and discussion 3.2 crystalline structure). No figure, No discussion. If some results have the specific peak or changes in diffraction pattern, authors must explain in detail.
Response 1: Thanks for your significant suggestions. We corrected this mistake and added the x-ray diffraction patterns as Figure.1. (Line 200)
Comment 2: Reduction of SGAPs by mild alkali treatment is benefical for normal amylose and waxy rice starches. What is the demerit of mild alkali modification? Although reviewer could understand the merit of this treatment, there were no explanation about demerit. It's better to add the texts.
Response 2: Thanks for your significant suggestions. We added some potential drawback in application of alkali treatment. The reduction of SGAPs by alkali reduced the granular stability, leading to the starch rupture more easily, especially for normal rice starch with HTC-11.3. It is undesirable for some food applications as a bulking agent, sorbent etc. (Line 317-319).
Comment 3: 3.3. Swelling power and solubility. Authors explained that the SP of normal amylose starch with HTC-11.3 increased (11.4 to 22.9), due to the great reduction of SGAPs content (0.76 to 0.23) in lines 206-208. In Table1, results of waxy rice showed much more reduction of SGAPs (0.96 to 0.23), however, the SP was slightly increased (27.7 to 33.3). What were the main reasons for these differences (relationships between SGAPs and SP) ? It is essential to add discussions with the references in 208.
Response 3: Thanks for your significant suggestions. We revised the manuscript according to the reviewer’s suggestions. The difference of SP between normal and waxy starches with HTC-11.3 might be related to a special type of SGAPs in normal starch, granule-bound-starch synthase (GBSS). GBSS is a protein responsible for the synthesis of amylose [37]. GBSS is tightly bound to starch compared to other SGAPs. It was reported that the removal of GBSS led to greater reduction of starch stability [38,39]. Therefore, when the reduction ratio of SGAPs was similar, the increase of SP of normal starch was greater than that of waxy starch (Line 216-224).
Comment 4: Line 280: please provide a clear comparison on how the presence of amylose influence the pasting properties compared to waxy starches under similar treatment conditions?
Response 4: Thanks for your significant suggestions. We added the influence of amylose on pasting properties of native normal starch, while the presence of amylose showed little influence on changes of pasting properties by alkali treatment. Specially, the normal rice starch with HTC-11.3 showed greater decrease of TV and FV, mainly due to the reduction of SGAPs (particular GBSS). The reduction of SGAPs significantly reduced the granular stability. Starch granules could swell and rupture more easily, resulting in the decrease of trough and final viscosity [39] (Line 281-282, 295-299).
Comment 5: Table 1 and Table 2 – please correct the significant figures
Response 5: Thanks for your significant suggestions. The Duncan’s tests were taken among the different treatment for a same cultivar. We revised the annotation below the table according to the reviewer’s suggestion. (Line 179-181, 276-278, 302-304)

Round 2
Reviewer 2 Report
Comments and Suggestions for Authors
Authors have revised the manuscript point by point according to the reviwer's comments. Some references were newly cited to support discussion.Otherwise, there are still short of explanation, therefore I raise some questions and comments as follows.
Major
The order of texts especially discussion is not suitable. For example, the result about thermal properties were shown in 3.4, however, in 3.1, authors showed the as follows: "The reduction of SGAPs induced starch to swell and gelatinize more easily, resulting in the changes of swelling, pasting and thermal properties (seen below)."
Ofcourse the obtained results were not explained from single experiment(analysis), therefore, it is difficult to explain the phenomena.
However, in 3.1, authors have shown about SAGPs content, so modify the text based on the result. I understand the complicated discusion cross-linked at any directions. For further better understanding, authors had better modify the discussion.
L171-175 It must be the common sense for food researchers."It might be because the higher temperature could increase the reactivity of alkali molecules [33], resulting in a significant reduction in protein content."
L173-175 Authors newly add the lines, however, what is the partial thermal-sensitive proteins in this present study?
As shown in the abstract (After mild alkali treatment, the 21 protein content of normal and waxy rice starches decreased from 0.76% to 0.23% and from 0.89% 22 to 0.23%, respectively.), these results were important point. Therefore, the explanation about how mild alkali treatment worked is one of the core discussion. It is essential for authors to add the explanation more in detail.
Especially, what is the partial thermal sensitive protein in normal rice and waxy rice ?
Figure 1, some alphabets were overwritten on the x-ray diffraction pattern lines. Move them to make them clear.
Title and scale of the vertical axis was not shown. In most cases for x-ray diffraction patterns,it might be the relative intensity or intensity.
Font size of horizontal axis (numbers and 2theta) were too small, thus increase the size.
Legend of figuer 1, Waxy and Normal should be waxy and normal.
Authors cited additional references and added the discussions.L192-198
Alkali could induce ionization of starch hydroxyl group by disrupting hydrogen bond of chains, causing reversible swelling of 195
starch granule by repulsion of anions [16,36]. Starch hydrogen groups were reinforced 196
by the hydrogen bond of double helical structure in crystalline regions during alkali 197
treatment. Therefore, no changes of starch RC occurred with alkali treatment.
In this present study, the originility exists in "MILD ALKALI" treatment, I guess. Therefore, when authors cited references about alkali treatment, the detailed condition (pH, temp etc) must be shown even in the discussion.
Authors add the discussion about swelling power (stability) with the modification of starch surface structure.
L218-
"The difference of SP between normal and waxy rice starches with HTC-11.3 might be related to a
special type of SGAPs in normal rice starch, granule-bound-starch synthase (GBSS).
GBSS is a protein responsible for the synthesis of amylose [37]. GBSS is tightly bound to starch
compared to other SGAPs. It was reported that the removal of GBSS led to greater reduction of starch stability [38,39]."
The explanation seems to be apparently correct, however, why authors think the removal of GBSS from the starch surface. Authors mentioned about SGAPs in lines 65 to 72 in introduction.
However, authors only focused on the physicochemial properties with the effects of SGAPs. It's better to show some protein examples (names or functions etc) including GBSS at the first appearance of SGAPs in introduction. If any more additional information, add them especially the state/composition of starch surface of these samples.
I wondar the removal mechanism based on what.... how to remove the GBSS from starch. Authors have better to add the explanation. Additionally, some researcher have investigated that treated starch maybe attributed to the formation of amylose-lipid complexes, which reduces granule swelling and causes leaching of amylose from the granules.
Authors mentioned about starch granule-associated proteins (SGAPs) in discussion, how did the lipid (amylose-lipid complex) effect on the changes in physicochemical properties ? Otherwise, ofcourse, if amylose-lipid complex was not investigated on the normal and waxy rice in this present study, authors need not to add the explanation above.
Minor
Abbreviations
Table 2 were shown with abbreviations from thermal analyses. In contrast, table 1 did not used abbreviations.
What was the difference with or without abbreviations ?
Comments on the Quality of English LanguageOver all, the quality of English language is good, however, some texts need modification.
Author Response
Comment 1:
Authors have revised the manuscript point by point according to the reviewer's comments. Some references were newly cited to support discussion. Otherwise, there are still short of explanation, therefore I raise some questions and comments as follows.
Response 1: Thanks for your significant suggestions. We revised the manuscript thoroughly according to the reviewer’s suggestions. Our responses are listed point-by-point as follows (changes shown in revised manuscript in red):
Major
Comment 2: The order of texts especially discussion is not suitable. For example, the result about thermal properties were shown in 3.4, however, in 3.1, authors showed the as follows: "The reduction of SGAPs induced starch to swell and gelatinize more easily, resulting in the changes of swelling, pasting and thermal properties (seen below)." Of course the obtained results were not explained from single experiment(analysis), therefore, it is difficult to explain the phenomena. However, in 3.1, authors have shown about SAGPs content, so modify the text based on the result. I understand the complicated discussion cross-linked at any directions. For further better understanding, authors had better modify the discussion.
Response 2: Thanks for your significant suggestions. We revised the manuscript and modified the order of some sentences according to the reviewer’s suggestions. We deleted the discussion "The reduction of SGAPs induced starch to swell and gelatinize more easily, resulting in the changes of swelling, pasting and thermal properties (seen below)." and added related information in Introduction (Line 67-78). “SGAPs are mainly comprised of proteins related to starch synthesis, e.g. starch synthase (SS), granule bound starch synthase (GBSS), 1,4-α-glucan-branching enzyme (GBE), and the residual seed storage proteins can be considered as SGAPs technologically. SGAPs can interact with starch molecules in various ways, including covalent bonding, hydrogen bonding, hydrophobic action, and van der Waal’s force [17], affecting the properties of starch [18,19]. The presence of SGAPs can increase starch stability, and inhibit starch swelling and gelatinization [17,20]. The removal of SGAPs weakens the granular structure of starch, resulting in the increase of swelling power and decrease of gelatinization temperature. Specially, GBSS, which is responsible for the synthesis of amylose, shows greater effect on starch properties than other proteins, as the removal of GBSS lead to specifically increase of breakdown of starch.” Besides, we revised the discussion in 3.4 (Line 273-278), “The presence of SGAPs can maintain the structure of starch granule and enhance the stability of starch granule [38]. It was reported that the removal of SGAPs can significantly decrease the gelatinization temperature of starches [39], it might be because SGAPs acts as a barrier to hinder water from entering starch granule, thus enhancing the thermal resistance of starch and further delaying the gelatinization [39].”
Comment 3: L171-175 It must be the common sense for food researchers. "It might be because the higher temperature could increase the reactivity of alkali molecules [33], resulting in a significant reduction in protein content."
Response 3: Thanks for your significant suggestions. We revised the sentence according to the reviewer’s suggestions. (Line 45).
Comment 4: L173-175 Authors newly add the lines, however, what is the partial thermal-sensitive proteins in this present study? As shown in the abstract (After mild alkali treatment, the protein content of normal and waxy rice starches decreased from 0.76% to 0.23% and from 0.89% to 0.23%, respectively.), these results were important point. Therefore, the explanation about how mild alkali treatment worked is one of the core discussion. It is essential for authors to add the explanation more in detail. Especially, what is the partial thermal sensitive protein in normal rice and waxy rice?
Response 4: Thanks for your significant suggestions. We revised the manuscript to better explain the changes of SGAPs according to the reviewer’s suggestion (Line 166-172, 184-192). For the SGAPs reduction in RTCs: “The reduction of protein contents for normal and waxy rice starches in RTCs was found to be 0.73% to 0.66% and 0.96% to 0.85%, respectively. The insolubility of rice proteins is mainly due to the presence of sulfhydryl groups from cysteine residues and amide bonds from glutamine and asparagine residues, which could form disulfide bonds and hydrogen bonds to maintain the protein structure [28]. The cleavage of disulfide bond and deamination of amide bonds could occur in alkali condition, increasing the solubility of SGAPs [8,29]. Besides, the starch granules swell in alkali, thus SGAPs could leach out more easily [27].” Then the great reduction of SGAPs contents, from 0.79% to 0.23% and from 0.94% to 0.23% respectively, could be due to the promotion of hydrothermal effect on alkali. “The higher temperature could increase the reactivity of alkali molecules [22], resulting in a significant reduction in protein content. It was reported that increase of temperature from 25°C to 55°C during maize starch extraction by 0.1% NaOH reduced the protein content of starch from 0.75% to 0.19% [30], similar to the result of this study. It might be because high temperature could promote of the destruction of disulfide bonds and the deamination reaction in alkali condition, thus alkali treatment could efficiently destroy the sulfhydryl groups and induce the deamination of glutamine and asparagine residues [29]. Therefore, the structure of SGAPs was destroyed and the solubility of SGAPs increased, resulting in the great reduction of protein content in HTC-11.3.” Then, for the thermal sensitive protein, as “There was no significant difference in the SGAPs content between native starch and their counterpart in HTCs”, it could be speculated that the individual mild thermal treatment had no effect on SGAPs content, it only could promote the destructive effect of alkali for SGAPs.
Comment 5: Figure 1, some alphabets were overwritten on the x-ray diffraction pattern lines. Move them to make them clear. Title and scale of the vertical axis was not shown. In most cases for x-ray diffraction patterns,it might be the relative intensity or intensity. Font size of horizontal axis (numbers and 2theta) were too small, thus increase the size. Legend of figuer 1, Waxy and Normal should be waxy and normal.
Response 5: Thanks for your significant suggestions. We revised Figure 1 according to the reviewer’s suggestions.
Comment 6: Authors cited additional references and added the discussions.L192-198. Alkali could induce ionization of starch hydroxyl group by disrupting hydrogen bond of chains, causing reversible swelling of starch granule by repulsion of anions [16,36]. Starch hydrogen groups were reinforced by the hydrogen bond of double helical structure in crystalline regions during alkali treatment. Therefore, no changes of starch RC occurred with alkali treatment.
In this present study, the originility exists in "MILD ALKALI" treatment, I guess. Therefore, when authors cited references about alkali treatment, the detailed condition (pH, temp etc) must be shown even in the discussion.
Response 6: Thanks for your significant suggestions. We revised the sentence according to the reviewer’s suggestions. (Line 208-218). The alkali treatment had no effect on the RC of starch, indicating that alkali treatment did not destroy the crystal structure, as the low concentration of alkali was insufficient to disturb the closely packed double helix. It was reported that 0.1M NaOH solution (pH 13, 25°C) induced the expansion of amorphous regions but had no significant effect on crystal regions [32]. And it was believed that alkali environment in chemical modifications (pH 8-12, 25°C -50°C) could induce ionization of starch hydroxyl group by disrupting hydrogen bond of chains, causing reversible swelling of starch granule by repulsion of anions [16,32]. Starch hydrogen groups were reinforced by the hydrogen bond of double helical structure in crystalline regions during alkali treatment. Therefore, no changes of starch RC occurred with alkali treatment, similar to the result of alkali-treated sago starch (pH 12.4, 25°C) [11].
Comment 7: Authors add the discussion about swelling power (stability) with the modification of starch surface structure.L218-"The difference of SP between normal and waxy rice starches with HTC-11.3 might be related to a special type of SGAPs in normal rice starch, granule-bound-starch synthase (GBSS). GBSS is a protein responsible for the synthesis of amylose [37]. GBSS is tightly bound to starch compared to other SGAPs. It was reported that the removal of GBSS led to greater reduction of starch stability [38,39]." The explanation seems to be apparently correct, however, why authors think the removal of GBSS from the starch surface. Authors mentioned about SGAPs in lines 65 to 72 in introduction. However, authors only focused on the physicochemial properties with the effects of SGAPs. It's better to show some protein examples (names or functions etc) including GBSS at the first appearance of SGAPs in introduction. If any more additional information, add them especially the state/composition of starch surface of these samples.
Response 7: Thanks for your significant suggestions. We revised the manuscript according to the reviewer’s suggestions. We added more information of SGAPs when they first appeared in introduction (Line 67-78). “SGAPs are mainly comprised of proteins related to starch synthesis, e.g. starch synthase (SS), granule bound starch synthase (GBSS), 1,4-α-glucan-branching enzyme (GBE), and the residual seed storage proteins can be considered as SGAPs technologically. SGAPs can interact with starch molecules in various ways, including covalent bonding, hydrogen bonding, hydrophobic action, and van der Waal’s force [17], affecting the properties of starch [18,19]. The presence of SGAPs can increase starch stability, and inhibit starch swelling and gelatinization [17,20]. The removal of SGAPs weakens the granular structure of starch, resulting in the increase of swelling power and de-crease of gelatinization temperature. Specially, GBSS, which is responsible for the syn-thesis of amylose, shows greater effect on starch properties than other proteins, as the removal of GBSS lead to specifically increase of breakdown of starch.”
As for composition of SGAPs, it was reported that the SGAPs of rice starch ranged from 10-109 kDa, with a main band at c.a. 60 kDa for GBSS (Han & Hamaker. Starch - Stärke 2002, 54, 454-460; Ye et al. Food Chem. 2019, 276, 754-760). Han and Hamaker reported that alkali (with pH 12.5 at room temperature for 24 h) significantly decrease the content of all SGAPs (Han & Hamaker. Starch - Stärke 2002, 54, 454-460). It also gives our team some inspiration, and we will work further on the state/composition of SGAPs of starch by the mild alkali treatment.
Comment 8: I wondar the removal mechanism based on what.... how to remove the GBSS from starch. Authors have better to add the explanation. Additionally, some researcher have investigated that treated starch maybe attributed to the formation of amylose-lipid complexes, which reduces granule swelling and causes leaching of amylose from the granules. Authors mentioned about starch granule-associated proteins (SGAPs) in discussion, how did the lipid (amylose-lipid complex) effect on the changes in physicochemical properties ? Otherwise, of course, if amylose-lipid complex was not investigated on the normal and waxy rice in this present study, authors need not to add the explanation above.
Response 8: Thanks for your significant suggestions. we added the explanation according to the reviewer’s suggestion (Line 166-172). “The insolubility of rice proteins is mainly due to the presence of sulfhydryl groups from cysteine residues and amide bonds from glutamine and asparagine residues, which could form disulfide bonds and hydrogen bonds to maintain the protein structure [28]. The cleavage of disulfide bond and deamination of amide bonds could occur in alkali condition, increasing the solubility of SGAPs [8,29]. Besides, the starch granules swell in alkali, thus SGAPs could leach out more easily [27].” As for lipid, we took a preliminary experiment to determine the lipid content during alkali treatment. The results showed that the lipid contents in rice starch was significantly less than protein contents, and there was no significant change during alkali treatment. For example, the lipid content of native rice starch and HTC-11.3 were 0.16±0.02% and 0.17±0.01%, respectively. Therefore, in this study we focused on the alkali effects on starch proteins but not lipids. But it is also a good inspiration for us and we will work further from this perspective.
Comment 9:
Minor
Abbreviations
Table 2 were shown with abbreviations from thermal analyses. In contrast, table 1 did not used abbreviations. What was the difference with or without abbreviations?
Response 9: Thanks for your significant suggestions. We revised Table 1 and use abbreviations (RC, SP, SOL) to replace “relative crystallinity, swell power and solubility” according to the reviewer’s suggestions. (Line 194).

Round 3
Reviewer 2 Report
Comments and Suggestions for Authors
Authors have totally revised the manuscript point by point according to the reviewer's comments including Figure and Table. Some references were newly cited to support discussion. Therefore, reviewer recommends accept as is.